# Approximating CNNs with Bag-of-local-Features models works surprisingly well on ImageNet

Wieland Brendel  and Matthias Bethge

Eberhard Karls University of Tübingen, Germany
Werner Reichardt Centre for Integrative Neuroscience, Tübingen, Germany
Bernstein Center for Computational Neuroscience, Tübingen, Germany
`{wieland.brendel, matthias.bethge}@bethgelab.org`

## Abstract

Deep Neural Networks (DNNs) excel on many complex perceptual tasks but it has proven notoriously difficult to understand how they reach their decisions. We here introduce a high-performance DNN architecture on ImageNet whose decisions are considerably easier to explain. Our model, a simple variant of the ResNet-50 architecture called BagNet, classifies an image based on the occurrences of small local image features without taking into account their spatial ordering. This strategy is closely related to the bag-of-feature (BoF) models popular before the onset of deep learning and reaches a surprisingly high accuracy on ImageNet ($87.6\%$ top-5 for $33 \times 33$ px features and Alexnet performance for $17 \times 17$ px features). The constraint on local features makes it straight-forward to analyse how exactly each part of the image influences the classification. Furthermore, the BagNets behave similar to state-of-the art deep neural networks such as VGG-16, ResNet-152 or DenseNet-169 in terms of feature sensitivity, error distribution and interactions between image parts. This suggests that the improvements of DNNs over previous bag-of-feature classifiers in the last few years is mostly achieved by better fine-tuning rather than by qualitatively different decision strategies.

## 1 Introduction

A big obstacle in understanding the decision-making of DNNs is due to the complex dependencies between input and hidden activations: for one, the effect of any part of the input on a hidden activation depends on the state of many other parts of the input. Likewise, the role of a hidden unit on downstream representations depends on the activity of many other units. This dependency makes it extremely difficult to understand how DNNs reach their decisions.

To circumvent this problem we here formulate a new DNN architecture that is easier to interpret *by design*. Our architecture is inspired by bag-of-feature (BoF) models which — alongside extensions such as VLAD encoding or Fisher Vectors — have been the most successful approaches to large-scale object recognition before the advent of deep learning (up to 75% top-5 on ImageNet) and classify images based on the counts, but not the spatial relationships, of a set of local image features. This structure makes the decisions of BoF models particularly easy to explain.

To be concise, throughout this manuscript the concept of *interpretability* refers to the way in which evidence from small image patches is integrated to reach an image-level decision. While basic BoF models perform just a simple and transparent spatial aggregate of the patch-wise evidences, DNNs non-linearly integrate information across the whole image.

In this paper we show that it is possible to combine the performance and flexibility of DNNs with the interpretability of BoF models, and that the resulting model family (called *BagNets*) is able to reach high accuracy on ImageNet even if limited to fairly small image patches. Given the simplicity of BoF models we imagine many use cases for which it can be desirable to trade a bit of accuracy for better interpretability, just as this is common e.g. for linear function approximation. This includes diagnosing failure cases (e.g. adversarial examples) or non-iid. settings (e.g. domain transfer),

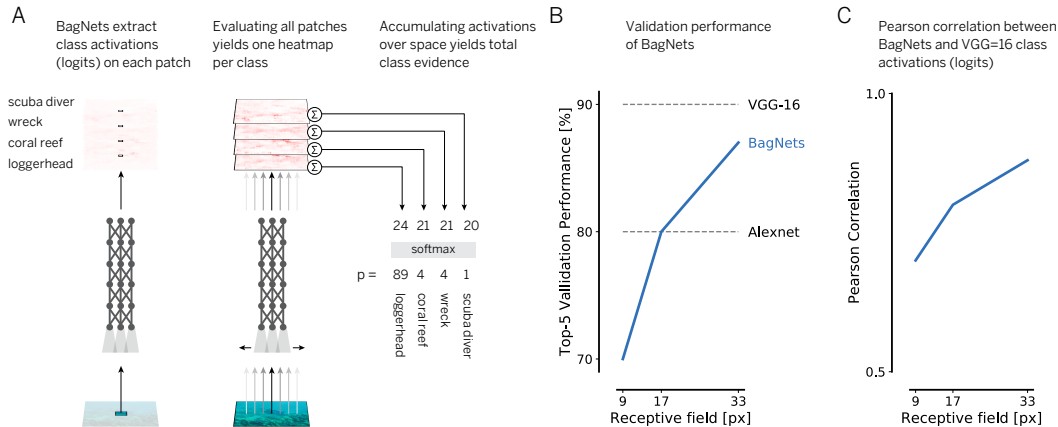

Figure 1: Deep bag-of-features models (BagNets). (A) The models extract features from small image patches which are each fed into a linear classifier yielding one logit heatmap per class. These heatmaps are averaged across space and passed through a softmax to get the final class probabilities. (B) Top-5 ImageNet performance over patch size. (C) Correlation with logits of VGG-16.

benchmarking diagnostic tools (e.g. attribution methods) or serving as interpretable parts of a computer vision pipeline (e.g. with a relational network on top of the local features).

In addition, we demonstrate similarities between the decision-making behaviour of BagNets and popular DNNs in computer vision. These similarities suggest that current network architectures base their decisions on a large number of relatively weak and local statistical regularities and are not sufficiently encouraged - either through their architecture, training procedure or task specification - to learn more holistic features that can better appreciate causal relationships between different parts of the image.

## 2 NETWORK ARCHITECTURE

We here recount the main elements of a classic bag-of-features model before introducing the simpler DNN-based BagNets in the next paragraph. Bag-of-feature representations can be described by analogy to bag-of-words representations. With bag-of-words, one counts the number of occurrences of words from a vocabulary in a document. This vocabulary contains important words (but not common ones like "and" or "the") and word clusters (i.e. semantically similar words like "gigantic" and "enormous" are subsumed). The counts of each word in the vocabulary are assembled as one long *term vector*. This is called the bag-of-words document representation because all ordering of the words is lost. Likewise, bag-of-feature representations are based on a vocabulary of visual words which represent clusters of local image features. The term vector for an image is then simply the number of occurrences of each visual word in the vocabulary. This term vector is used as an input to a classifier (e.g. SVM or MLP). Many successful image classification models have been based on this pipeline (Csurka et al., 2004; Jurie & Triggs, 2005; Zhang et al., 2007; Lazebnik et al., 2006), see O'Hara & Draper (2011) for an up-to-date overview.

BoF models are easy to interpret if the classifier on top of the term vector is linear. In this case the influence of a given part of the input on the classifier is independent of the rest of the input.

Based on this insight we construct a linear DNN-based BoF model as follows (see Figure 1): first, we infer a 2048 dimensional feature representation from each image patch of size q × q pixels using multiple stacked ResNet blocks and apply a linear classifier to infer the class evidence for each patch (*heatmaps*). We average the class evidence across all patches to infer the image-level class evidence (logits). This structure differs from other ResNets (He et al., 2015) only in the replacement of many 3 × 3 by 1 × 1 convolutions, thereby limiting the receptive field size of the topmost convolutional layer to q × q pixels (see Appendix for details). There is no explicit assignment to visual words. This could be added through a sparse projection into a high-dimensional embedding but we did not see benefits for interpretability. We denote the resulting architecture as BagNet-q and test $q \in [9, 17, 33]$.

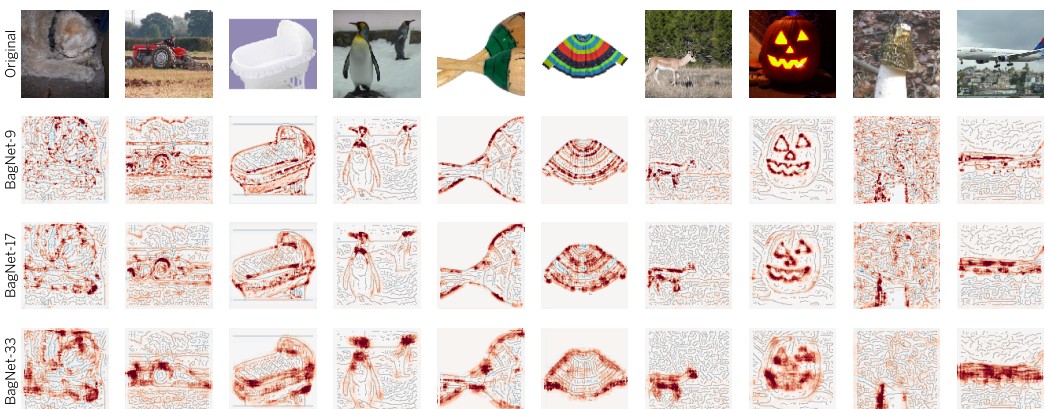

Figure 2: Heatmaps showing the class evidence extracted from of each part of the image. The spatial sum over the evidence is the total class evidence.

Note that an important ingredient of our model is the *linear* classifier on top of the local feature representation. The word *linear* here refers to the combination of a linear spatial aggregation (a simple average) and a linear classifier on top of the aggregated features. The fact that the classifier and the spatial aggregation are both linear and thus interchangeable allows us to pinpoint exactly how evidence from local image patches is integrated into one image-level decision.

## 3 RELATED LITERATURE

**BoF models and DNNs** There are some model architectures that fuse elements from DNNs and BoF models. Predominantly, DNNs were used to replace the previously hand-tuned feature extraction stage in BoF models, often using intermediate or higher layer features of pretrained DNNs (Feng et al., 2017; Gong et al., 2014; Ng et al., 2015; Mohedano et al., 2016; Cao et al., 2017; Khan et al., 2016) for tasks such as image retrieval or geographical scene classification. Other work has explored how well insights from DNN training (e.g. data augmentation) transfer to the training of BoF and Improved Fisher Vector models (Chatfield et al., 2014) and how SIFT and CNN feature descriptions perform (Babenko & Lempitsky, 2015). In contrast, our proposed BoF model architecture is simpler and closer to standard DNNs used for object recognition while still maintaining the interpretability of linear BoF models with local features. Furthermore, to our knowledge this is the first work that explores the relationship between the decision-making of BoF and DNN models.

**Interpretable DNNs** Our work is most closely related to approaches that use DNNs in conjunction with more interpretable elements. Pinheiro & Collobert (2014) adds explicit labelling of single pixels before the aggregation to an image-level label. The label of each pixel, however, is still inferred from the whole image, making the pixel assignments difficult to interpret. Xiao et al. (2015) proposed a multi-step approach combining object-, part- and domain-detectors to reach a classification decision. In this process object-relevant patches of variable sizes are extracted. In contrast, our approach is much simpler, reaches higher accuracy and is easier to interpret. Besides pixel-level attention-based mechanisms there are several attempts to make the evidence accumulation more interpretable. Hinton et al. (2015) introduced soft decision trees that are trained on the predictions of neural networks. While this increases performance of decision trees, the gap to neural networks on data sets like ImageNet is still large. In Li et al. (2017) an autoencoder architecture is combined with a shallow classifier based on prototype representations. Chen et al. (2018) uses a similar approach but is based on a convolutional architecture to extract class-specific prototype patches. The interpretation of the prototype-based classification, however, is difficult because only the L2 norm between the prototypes and the extracted latent representations is considered[1]. Finally, the class activation maps by Zhou et al. (2015) share similarities to our approach as they also use a CNN with global average pooling and a linear classifier in order to extract class-specific heatmaps. However, their latent representations are

---

[1]Just because two images have a similar latent representation does not mean that they share any similarities in terms of human-interpretable features.

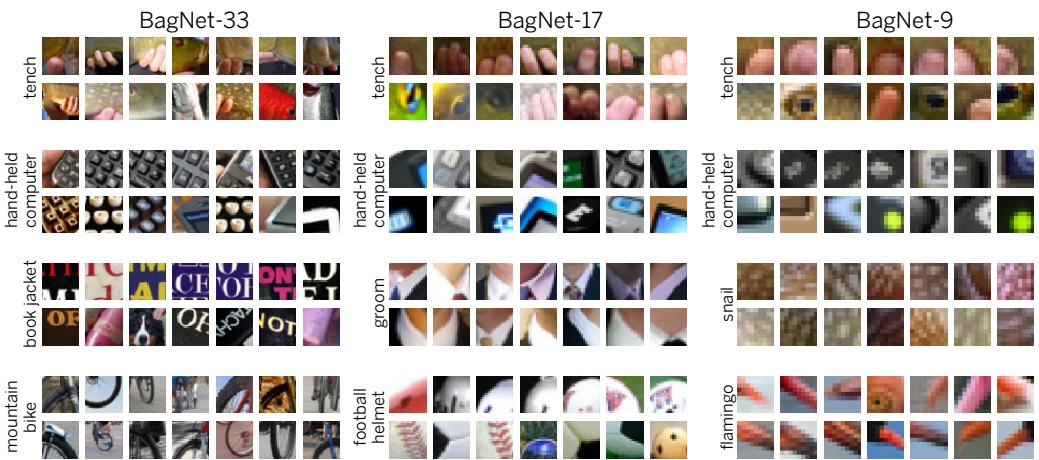

Figure 3: Most informative image patches for BagNets. For each class (row) and each model (column) we plot two subrows: in the top subrow we show patches that caused the highest logit outputs for the given class across all validation images with that label. Patches in the bottom subrow are selected in the same way but from all validation images with a *different* label (highlighting errors).

extracted from the whole image and it is unclear how the heatmaps in the latent space are related to the pixel space. In our approach the CNN representations are restricted to very small image patches, making it possible to trace exactly how each image patch contributes the final decision.

**Scattering networks**  Another related work by Oyallon et al. (2017) uses a scattering network with small receptive fields (14 x 14 pixels) in conjunction with a two-layer Multilayer Perceptron or a ResNet-10 on top of the scattering network. This approach reduces the overall depth of the model compared to ResNets (with matched classification accuracy) but does not increase interpretability (because of the non-linear classifier on top of the local scattering features).

A set of superficially similar but unrelated approaches are region proposal models (Wei et al., 2016; Tang et al., 2017; 2016; Arandjelovic et al., 2015). Such models typically use the whole image to infer smaller image regions with relevant objects. These regions are then used to extract a spatially aligned subset of features from the highest DNN layer (so information is still integrated far beyond the proposed image patch). Our approach does not rely on region proposals and extracts features only from small local regions.

## 4 RESULTS

In the first two subsections we investigate the classification performance of BagNets for different patch sizes and demonstrate insights we can derive from its interpretable structure. Thereafter we compare the behaviour of BagNets with several widely used high-performance DNNs (e.g. VGG-16, ResNet-50, DenseNet-169) and show evidence that their decision-making shares many similarities.

### 4.1 ACCURACY & RUNTIME OF BAGNETS ON IMAGENET

We train the BagNets directly on ImageNet (see Appendix for details). Surprisingly, patch sizes as small as $17 \times 17$ pixels suffice to reach AlexNet (Krizhevsky et al., 2012) performance (80.5% top-5 performance) while patches sizes $33 \times 33$ pixels suffice to reach close to 87.6%.

We also compare the runtime of BagNet-q ($q = 33, 17, 9$) in inference mode with images of size $3 \times 224 \times 224$ and batch size 64 against a vanilla ResNet-50. Across all receptive field sizes BagNets reach around 155 images/s for BagNets compared to 570 images/s for ResNet-50. The difference in runtime can be attributed to the reduced amount of downsampling in BagNets compared to ResNet-50.

### 4.2 EXPLAINING DECISIONS

For each q × q patch the model infers evidence for each ImageNet classes, thus yielding a high-resolution and very precise heatmap that shows which parts of the image contributes most to certain decisions. We display these heatmaps for the predicted class for ten randomly chosen test images in

Figure 2. Clearly, most evidence lies around the shapes of objects (e.g. the crip or the paddles) or certain predictive image features like the glowing borders of the pumpkin. Also, for animals eyes or legs are important. It's also notable that background features (like the forest in the deer image) are pretty much ignored by the BagNets.

Next we pick a class and run the BagNets across all validation images to find patches with the most class evidence. Some of these patches are taken from images of that class (i.e. they carry "correct" evidence) while other patches are from images of another class (i.e. these patches can lead to misclassifications). In Figure 3 we show the top-7 patches from both correct and incorrect images for several classes (rows) and different BagNets (columns). This visualisation yields many insights: for example, book jackets are identified mainly by the text on the cover, leading to confusion with other text on t-shirts or websites. Similarly, keys of a typewriter are often interpreted as evidence for handheld computers. The tench class, a large fish species, is often identified by fingers on front of a greenish background. Closer inspection revealed that tench images typically feature the fish hold up like a trophy, thus making the hand and fingers holding it a very predictive image feature. Flamingos are detected by their beaks, which makes them easy to confuse with other birds like storks, while grooms are primarily identified by the transition from suit to neck, an image feature present in many other classes.

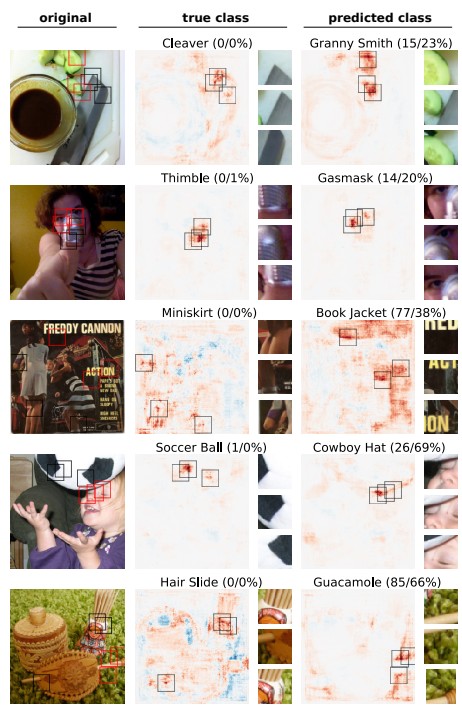

Figure 4: Images misclassified by BagNet-33 and VGG-16 with heatmaps for true and predicted label and the most predictive image patches. Class probability reported for BagNet-33 (left) and VGG (right).

In Figure 4 we analyse images misclassified by both BagNet-33 and VGG-16. In the first example the ground-truth class "cleaver" was confused with "granny smith" because of the green cucumber at the top of the image. Looking at the three most predictive patches plotted alongside each heatmap, which show the apple-like edges of the green cucumber pieces, this choice looks comprehensible. Similarly, the local patches in the "thimble" image resemble a gas mask if viewed in isolation. The letters in the "miniskirt" image are very salient, thus leading to the "book jacket" prediction while in the last image the green blanket features a glucamole-like texture.

### 4.3 Comparing the decision-making of BagNets and high-performance DNNs

In the next paragraphs we investigate how similar the decision-making of BagNets is to high-performance DNNs like VGG-16, ResNet-50, ResNet-152 and DenseNet-169. There is no single answer or number, partially because we lack a sensible distance metric between networks. One can

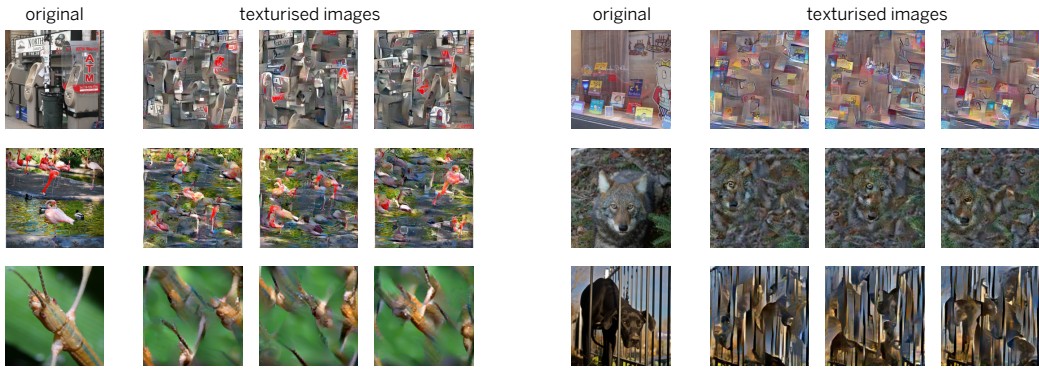

Figure 5: Examples of original and texturised images. A vanilla VGG-16 still reaches high accuracy on the texturised images while humans suffer greatly from the loss of global shapes in many images.

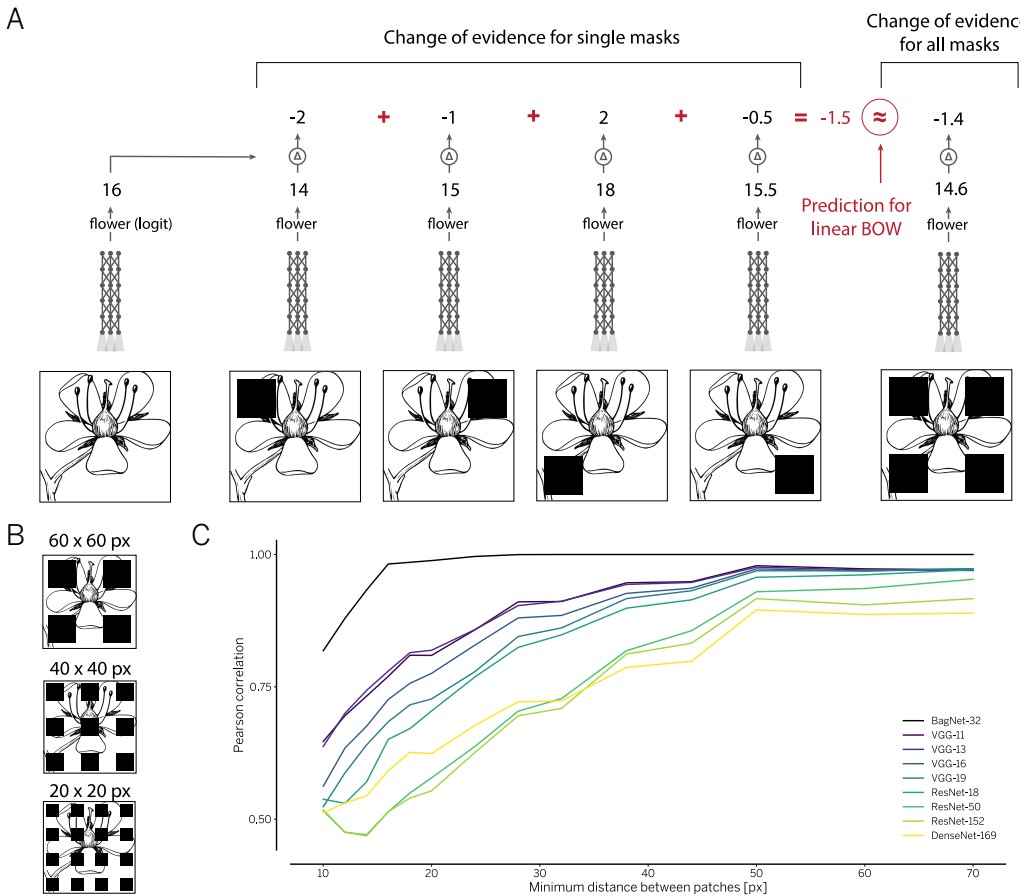

Figure 6: Interaction of spatially separated image parts. (A) Changes in class-evidence when single image patches are masked (centre) versus change when all patches are masked simultaneously (right). For linear BoF models both terms are the same. (B) Masking regions for different patch sizes. (C) Correlation between both terms for different DNNs over different patch sizes. Interactions are greatly depressed for image features larger than $30 \times 30$ px.

compare the pearson correlation between logits (for VGG-16, BagNet-9/17/33 reach 0.70 / 0.79 / 0.88 respectively, see Figure 1C), but this number can only give a first hint as it does not investigate the specific process that led to the decision. However, the decision-making of BagNets does feature certain key characteristics that we can compare to other models.

**Image Scrambling**   One core component of the bag-of-feature networks is the neglect of the spatial relationships between image parts. In other words, scrambling the parts across the image while keeping their counts constant does not change the model decision. Is the same true for current computer vision models like VGG or ResNets? Unfortunately, due to the overlapping receptive fields it is generally not straight-forward to scramble an image in a way that leaves the feature histograms invariant. For VGG-16 an algorithm that comes close to this objective is the popular texture synthesis algorithm based on the Gram features of the hidden layer activations (Gatys et al., 2015), Figure 5. For humans, the scrambling severely increases the difficulty of the task while the performance of VGG-16 is little affected (90.1% on clean versus 79.4% on texturised image).

This suggests that VGG, in stark contrast to humans, does not rely on global shape integration for perceptual discrimination but rather on statistical regularities in the histogram of local image features. It is well known by practioners that the aforementioned texture synthesis algorithm does not work for ResNet- and DenseNet architectures, the reasons of which are not yet fully understood.

**Spatially distinct image manipulations do not interact**   For BoF models with a linear (but not non-linear!) classifier we do not only expect invariance to the spatial arrangement of image parts, but also that the marginal presence or absence of an image part always has the same effect on the

evidence accumulation (i.e. is independent of the rest of the image). In other words, for a BoF model an image with five unconnected wheels (and nothing else) would carry more evidence for class "bike" than a regular photo of a bicycle; a linear BoF model simply ignores whether there is also a frame and a saddle. More precisely, let $\ell_{\text{model}}(\mathbf{x})$ be the class evidence (logit) as a function of the input $\mathbf{x}$ and let $\boldsymbol{\delta}_i$ be spatially separated and non-overlapping input modifications. For a BagNet-q it holds that

$$\ell_{\text{model}}(\mathbf{x}) - \ell_{\text{model}}(\mathbf{x} + \sum_i \boldsymbol{\delta}_i) = \sum_i \left( \ell_{\text{model}}(\mathbf{x}) - \ell_{\text{model}}(\mathbf{x} + \boldsymbol{\delta}_i) \right), \tag{1}$$

as long as the modifications are separated by more than $q$ pixels. We use the Pearson correlation between the LHS and RHS of eq. (1) as a measure of non-linear interactions between image parts. In our experiments we partition the image into a grid of non-overlapping square-sized patches with patch size $q$. We then replace every second patch in every second row (see Figure 6B) with its DC component (the spatial channel average) both in isolation (RHS of eq. (1)) and in combination (LHS of eq. (1)), see Figure 6A. This ensures that the masked patches are spaced by $q$ pixels and that always around 1/4 of the image is masked. Since most objects fill much of the image, we can expect that the masking will remove many class-predictive image features. We measure the Pearson correlation between the LHS and RHS of eq. 1 for different patch sizes $q$ and DNN models (Figure 6C). The results (Figure 6C) show that VGG-16 exhibits few interactions between image parts spaced by more than 30 pixels. The interactions increase for deeper and more performant architectures.

**Error distribution**   In Figure 7 we plot the top-5 accuracy within each ImageNet class of BagNet-33 against the accuracy of regular DNNs. For comparison we also plot VGG-11 against VGG-16. The analysis reveals that the error distribution is fairly consistent between models.

**Spatial sensitivity**   To see whether BagNets and DNNs use similar image parts for image classification we follow Zintgraf et al. (2017) and test how the prediction of DNNs is changing when we mask the most predictive image parts. In Figure 8 (top) we compare the decrease in predicted class probability for an increasing number of masked $8 \times 8$ patches. The masking locations are determined by the heatmaps of BagNets which we compare against random maskings as well as several popular attribution techniques (Baehrens et al., 2010; Sundararajan et al., 2017; Kindermans et al., 2018; Shrikumar et al., 2017) (we use the implementations of DeepExplain (Ancona et al., 2017)) which compute heatmaps directly in the tested models. Notice that these attribution methods have an advantage because they compute heatmaps knowing everything about the models (white-box setting). Nonetheless, the heatmaps from BagNets turn out to be more predictive for class-relevant image parts (see also Table 1). In other words, image parts that are relevant to BagNets are similarly relevant for the classification of normal DNNs. VGG-16 is most affected by the masking of local patches while deeper and more performant architectures are more robust to the relatively small masks, which again suggests that deeper architectures take into account larger spatial relationships.

## 5   DISCUSSION & OUTLOOK

In this paper we introduced and analysed a novel interpretable DNN architecture — coined BagNets — that classifies images based on linear bag-of-local-features representations. The results demonstrate that even complex perceptual tasks like ImageNet can be solved just based on small image features and without any notion of spatial relationships. In addition we showed that the key properties of BagNets, in particlar invariance to spatial relationships as well as weak interactions between image features, are also present to varying degrees in many common computer vision models like ResNet-50

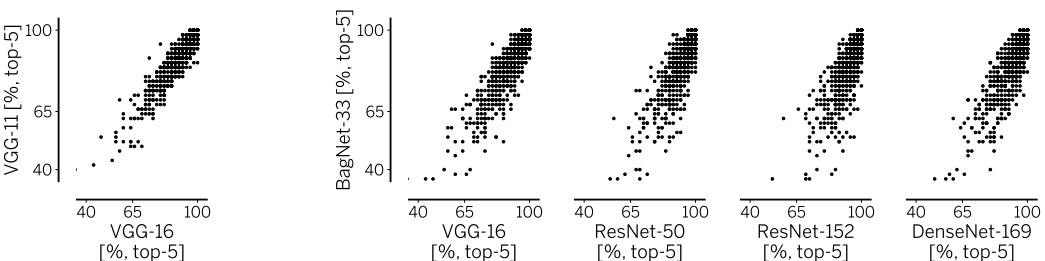

Figure 7: Scatter plots of class-conditional top-5 errors for different models.

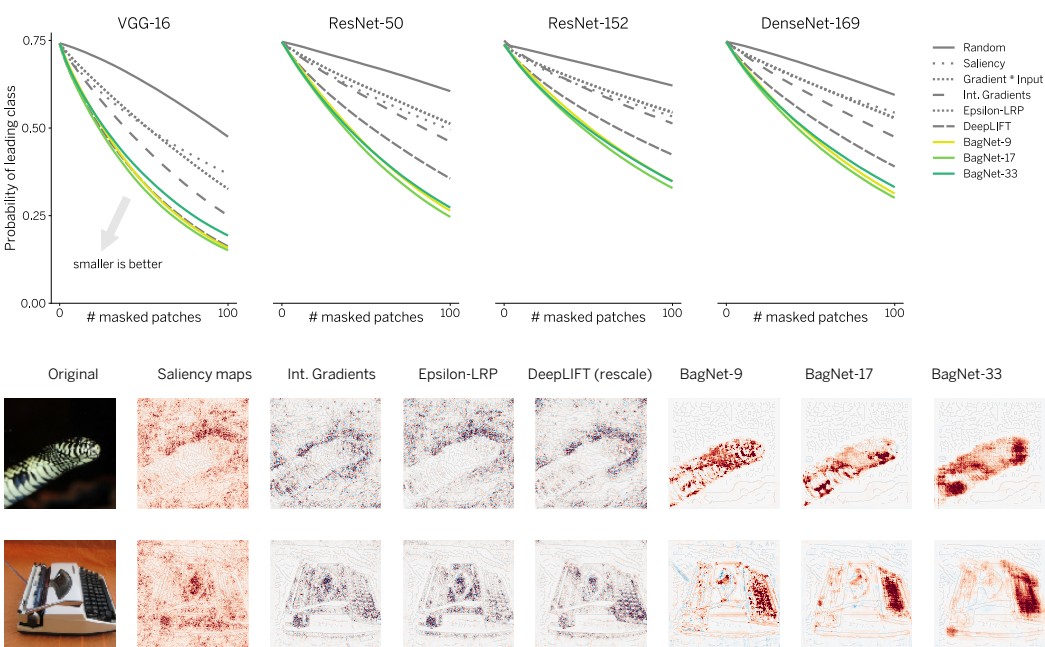

Figure 8: Similarity of image features used for object classification. (Top) Decrease of leading class probability in VGG-16, ResNet-50, ResNet-152 and DenseNet-169 if increasingly more patches are masked according to the heatmaps of BagNets and several popular attribution methods. The faster the decrease the more closely does the heatmap highlight image parts relevant for the model decisions making. Image parts relevant to the BagNets turn out to be similarly relevant for all models and outperform post-hoc attribution methods. (Bottom) The first four heatmaps show attributions computed on VGG-16, the other three heatmaps show the class evidence of BagNets.

| | Sali-ency | Int. Grad. | $\epsilon$-LRP | Deep LIFT | BN-9 | BN-17 | BN-33 |
|---|---|---|---|---|---|---|---|
| | white-box | | | | black-box | | |
| VGG-16 | 0.369 | 0.250 | 0.326 | 0.162 | 0.158 | **0.151** | 0.193 |
| ResNet-50 | 0.528 | 0.492 | 0.545 | 0.379 | 0.281 | **0.263** | 0.291 |
| ResNet-152 | 0.602 | 0.580 | 0.614 | 0.479 | 0.394 | **0.371** | 0.393 |
| DenseNet-169 | 0.589 | 0.515 | 0.571 | 0.423 | 0.339 | **0.326** | 0.359 |

Table 1: Average probability of leading class after masking the 100 patches ($8 \times 8$ pixels) with the highest attribution according to different heatmaps (columns).

or VGG-16, suggesting that the decision-making of many DNNs trained on ImageNet follows at least in part a similar bag-of-feature strategy. In contrast to the perceived "leap" in performance from bag-of-feature models to deep neural networks, the representations learnt by DNNs may in the end still be similar to the pre-deep learning era.

VGG-16 is particularly close to bag-of-feature models, as demonstrated by the weak interactions (Figure 6) and the sensitivity to the same small image patches as BagNets (Figure 8). Deeper networks, on the other hand, exhibit stronger nonlinear interactions between image parts and are less sensitive to local maskings. This might explain why texturisation (Figure 5) works well in VGG-16 but fails for ResNet- and DenseNet-architectures.

Clearly, ImageNet alone is not sufficient to force DNNs to learn more physical and causal representation of the world — simply because such a representation is not necessary to solve the task (local image features are enough). This might explain why DNNs generalise poorly to distribution shifts: a DNN trained on natural images has learnt to recognize the textures and local image features

associated with different objects (like the fur and eyes of a cat or the keys of a typewriter) and will inevitably fail if presented with cartoon-like images as they lack the key local image features upon which it bases its decisions.

One way forward is to define novel tasks that cannot be solved using local statistical regularities. Here the BagNets can serve as a way to evaluate a lower-bound on the task performance as a function of the observable length-scales. Furthermore, BagNets can be an interesting tool in any application in which it is desirable to trade some accuracy for better interpretability. For example, BagNets can make it much easier to spot the relevant spatial locations and image features that are predictive of certain diseases in medical imaging. Likewise, they can serve as diagnostic tools to benchmark feature attribution techniques since ground-truth attributions are directly available. BagNets can also serve as interpretable parts of a larger computer vision pipeline (e.g. in autonomous cars) as they make it easier to understand edge and failure cases. We released the pretrained BagNets (BagNet-9, BagNet-17 and BagNet-33) for PyTorch and Keras at `https://github.com/wielandbrendel/bag-of-local-features-models`.

Taken together, DNNs might be more powerful than previous hand-tuned bag-of-feature algorithms in discovering weak statistical regularities, but that does not necessarily mean that they learn substantially different representations. We hope that this work will encourage and inspire future work to adapt tasks, architectures and training algorithms to encourage models to learn more causal models of the world.

### ACKNOWLEDGMENTS

This work has been funded, in part, by the German Research Foundation (DFG CRC 1233 on Robust Vision) as well as by the Intelligence Advanced Research Projects Activity (IARPA) via Department of Interior / Interior Business Center (DoI/IBC) contract number D16PC00003.

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

## A  APPENDIX

The architecture of the BagNets is detailed in Figure A.1. Training of the models was performed in PyTorch using the default ImageNet training script of Torchvision (https://github.com/pytorch/vision, commit 8a4786a) with default parameters. In brief, we used SGD with momentum (0.9), a batchsize of 256 and an initial learning rate of 0.01 which we decreased by a factor of 10 every 30 epochs. Images were resized to 256 pixels (shortest side) after which we extracted a random crop of size $224 \times 224$ pixels.

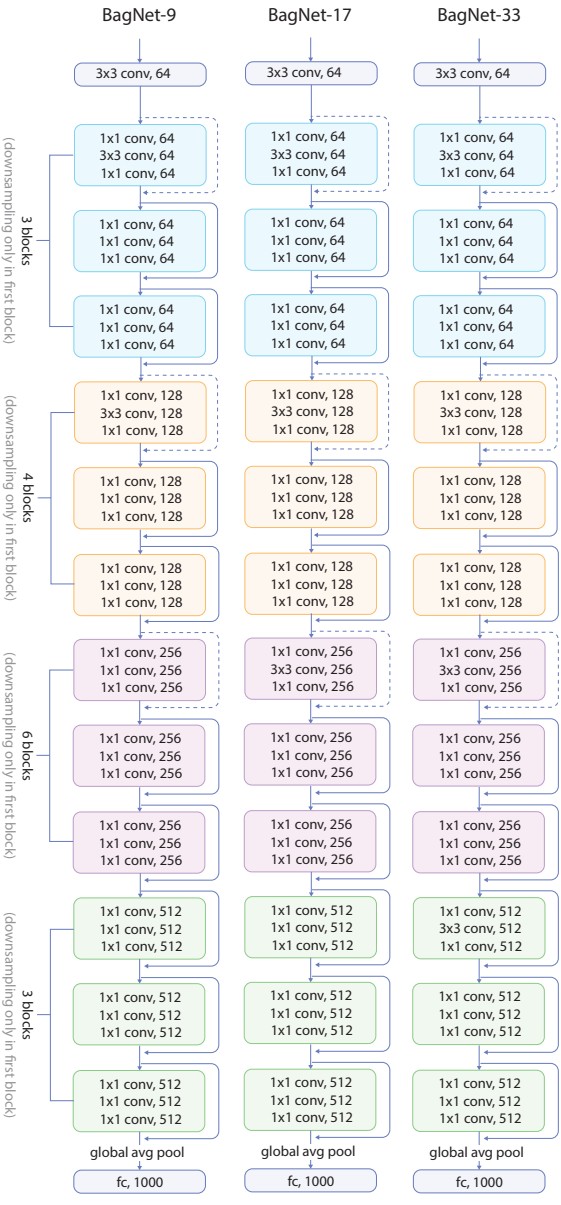

Figure A.1: The BagNet architecture is almost equivalent to the ResNet-50 architectures except for a few changes in the strides and the replacement of most $3 \times 3$ convolutions with $1 \times 1$ convolutions. Each ResNet block has an expansion of size four (that means the number of output feature maps is four times the number of feature maps within the block). The downsampling operation (dashed arrows) is a simple $1 \times 1$ convolution with stride 2.

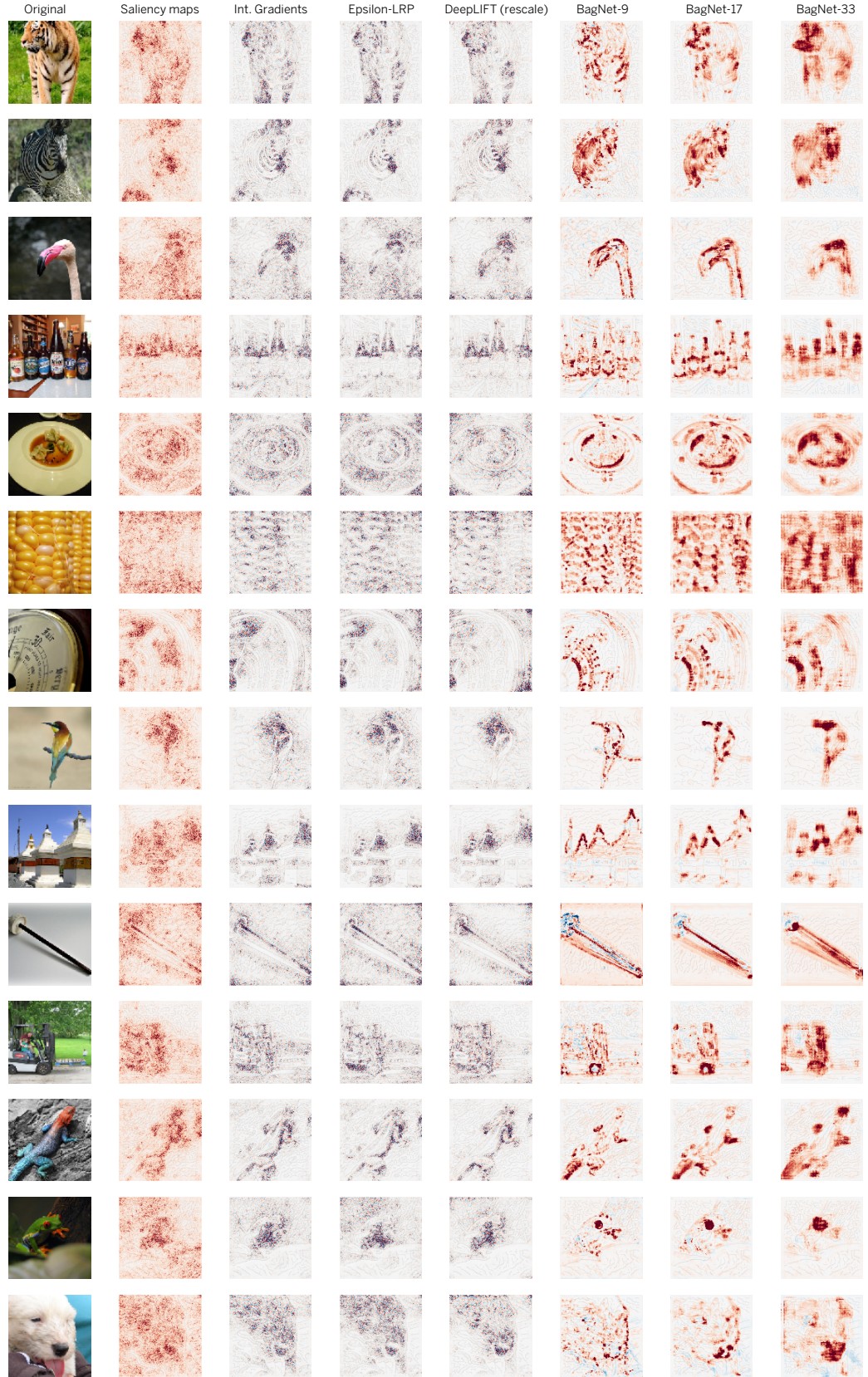

Figure A.2: Feature attributions of VGG generated using different methods (Saliency, Integrated Gradients, $\epsilon$-LRP and DeepLIFT) and feature attributions of BagNets.

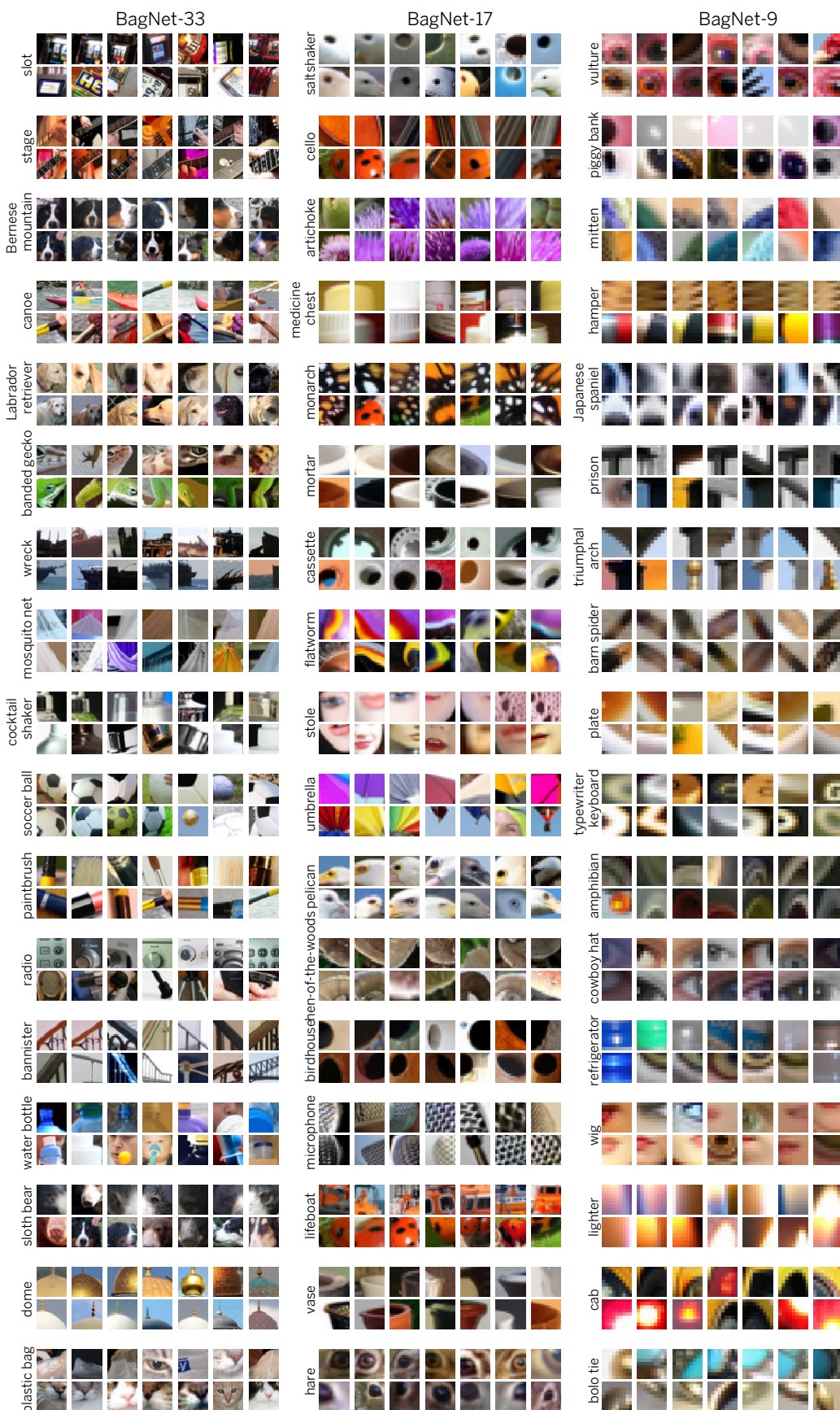

Figure A.3: Same as Figure 3 but for more classes.

## A.1 EFFECT OF LOGIT THRESHOLDING

We tested how sensitive the classification accuracy of BagNet-33 is with respect to the exact values of the logits for each patch. To this end we thresholded the logits in two ways: first, by setting all values *below* the threshold to the threshold (and all values above the threshold stay as is). In the second case we binarized the heatmaps by setting all values below the threshold to zero and all values above the threshold to one (this completely removes the amplitude). The results can be found in Figure A.4. Most interestingly, for certain binarization thresholds the top-5 accuracy is within 3-4% of the vanilla BagNet performance. This indicates that the amplitude of the heatmaps is not decisive.

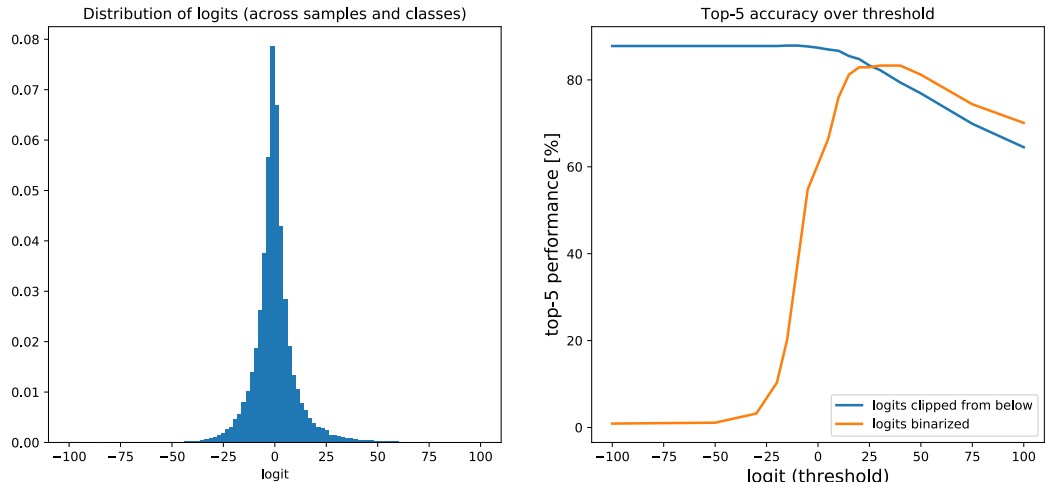

Figure A.4: Effect of thresholding the logits on the model performance (top-5 accuracy).

