# OpenReview forum: "Approximating CNNs with Bag-of-local-Features models works surprisingly well on ImageNet"
_ICLR.cc/2019/Conference_

### Official Review · AnonReviewer1 · 2018-11-01
**Interesting empirical analysis**

**Rating:** 7
**Confidence:** 4

**Review:**

This is an experimental paper that investigates how spatial ordering of patches influences the classification performances of CNNs. To do so, the authors design CNNs close to ResNets that almost only consist in a simple cascade of 1x1 convolutions, obtaining relatively small receptive field. It is an interesting read, and I recommend it as a valuable contribution to ICLR, that might lead to nice future works.

I have however several comments and questions, that I would like to be addressed.

1) First I think a reference is missing. Indeed, to my knowledge, it is not the first work to use this kind of techniques. Cf [1]. This does not alterate however the novelty of the approach.

2) « We construct a linear DNN-based BoF » : I do not like this formulation. Here, you assume that you build a ResNet-50 with 1x1 as a representation and have a last final linear layer as a classifier. One could also claim it is a ResNet-48 as a representation followed by 2 layers of 1x1 as a classifier.

3) « our proposed model architecture is simpler » this is very subjective because for instance the FV models are learned in a layer-wise fashion, which makes their learning procedure more interpretable because each layer objective is specified. Furthermore, analyzing these models is now equivalent to analyze a cascade of fully connected layers, which is not simple at all.

4) Again, the interpretability mentioned in Sec. 3  is in term of spatial localization, not mapping. I think it is important to make clear this consideration. Indeed, this work basically leaves the problem of understanding general CNNs to the problem of understanding MLPs.

5) The graphic of the Appendix A is a bit misleading : it seems 13 downsampling are performed whereas it is not the case, because the first element of each group of block is actually only done once.(if I understood correctly)

6) I think the word feature is sometimes mis-used: sometimes it seems it can refer to a patch, sometimes to the code for a patch. (« Surprisingly, feature sizes assmall as 17 × 17 pixels »)

I got also few questions:
Q1 : I was wondering if you did try manifold learning on the patches ? Do you expect it to work ?
Q2 : Is there a batch normalization in the FC or a normalization? Did you try to threshold the heat maps before feeding them to the linear layer? I'm wondering indeed if the amplitude of those heatmaps is really key.
Q3 : do you think it would be easy to exploit the non-overlapping patches for a better parallelization of computations ?

Finally, I find very positive the amount of experiments to test the similarity with standard CNNs. Of course, it’s far from being a formal proof, but I think it is a very nice first step.

[1] Oyallon, Edouard, Eugene Belilovsky, and Sergey Zagoruyko. "Scaling the scattering transform: Deep hybrid networks." International Conference on Computer Vision (ICCV). 2017.

---

> ### Comment · AnonReviewer1 · 2018-11-22
> **Rebuttal?**
>
> I'm respectfully wondering if the authors had any thoughts w.r.t. this review?

---

> > ### Author Response · Authors · 2018-11-22
> > **Thanks for the insightful review!**
> >
> > Please excuse our late response and thanks for your insightful feedback and your positive assessment! We are in the middle of the thresholding experiment you suggested. We report on the results later today or tomorrow and would like to respond to the rest of your comments and suggestions:
> >
> > 1) Reference to scattering network [1]
> > That’s indeed an interesting and related reference with respect to spatially constrained network architectures that we’ll add to the manuscript. The scattering network extracts patches of size 14 x 14 pixels and employs an additional two-layer fully connected network (or a ResNet-10) on top of those features. That unfortunately makes the spatial aggregation again harder to pinpoint (compared to BagNets) but the work is nonetheless an important precursor. We'll add the reference to the manuscript.
> >
> > 2) « We construct a linear DNN-based BoF »
> > Thanks for pointing this out, we have to sharpen the related definitions. For us, the *classifier* is the part of the model _after_ the spatial aggregation (spatial average). The distinction of a linear vs a non-linear classifier is important because only the first is interchangeable with the spatial aggregation, which means that we can interpret the model as extracting evidence from each patch, which is then averaged across all patches. This guarantees that we can exactly attribute how important each patch is for the final model decision. This is not true if the classifier is non-linear. We will clarify this definition and its importance in the manuscript and hope that this sufficiently addresses  your concern.
> >
> > 3) simplicity is subjective
> > We agree that one should add a more specific qualifier on “simpler” and “more interpretable” as these terms can refer to different concepts. Thanks for pointing this out. What we mean is that the decision making process is more transparent: whereas CNNs take the whole image and somehow churn out a final label, the BagNets linearly accumulate evidence from very small patches to make a decision. This restricts the decision making to features of small size (e.g. BagNets cannot use shape), which makes it easier to understand how certain decisions have been reached (in terms of which image parts have been used). We will clarify this in the manuscript.
> >
> > 4) “this work basically leaves the problem of understanding general CNNs to the problem of understanding MLPs”
> > In terms of the evidence extraction from local patches you are right: our paper has nothing to add here. However, from the perspective of the whole image the decision making is more transparent (in terms of which image features are being used for classification) than for general CNNs. In addition, the MLPs now only look at small image patches, which means that the input distribution has much lower entropy. This makes it potentially easier to analyse what evidence is extracted by the MLPs. We will add a clarifying sentence into the manuscript.
> >
> > 5) Network depiction in Appendix A
> > Thanks for pointing this out, indeed it looks as if the down pooling is performed in each block whereas in reality it’s only used in the first. We’ll add an arrow with a description saying “downsampling is only performed in the first block” in each group.
> >
> > 6) Usage of word “features”
> > Thanks for pointing this out, we’ll go through the manuscript and change the usage to “image feature” (for small patches) or “feature embedding” (to refer to the feature vector extracted by the DNN).
> >
> > Q1 : I was wondering if you did try manifold learning on the patches ? Do you expect it to work ?
> > That’s an excellent question that we have not yet explored. We have some ideas in that direction (unsupervised learning of low-level image features) but no results to support or refute this idea.
> >
> > Q2 : Is there a batch normalization in the FC or a normalization? Did you try to threshold the heat maps before feeding them to the linear layer? I'm wondering indeed if the amplitude of those heatmaps is really key.
> >
> > There is no normalisation between the average pooling and the linear classifier. We report back on the thresholding experiment today or tomorrow.
> >
> > Q3 : do you think it would be easy to exploit the non-overlapping patches for a better parallelization of computations ?
> >
> > That’s a good point, indeed that would be possible. One could simply cut an image into smaller parts (e.g. into 128 x 128 patches that overlap by q pixels on the borders where q is the RF size of the BagNet), then run each through the BagNets on separate GPUs and then sum the output of the linear classifier (before the softmax) from each part/GPU.

---

> > > ### Author Response · Authors · 2018-11-22
> > > **Thresholding has little effect on accuracy**
> > >
> > > As promised we tested how thresholding the logits affects accuracy. We thresholded in two ways: first, by setting all values *below* the threshold to the threshold (and all values above the threshold stay as is). In the second case we binarised the heatmaps by setting all values below the threshold to zero and all values above the threshold to one (this completely removes the amplitude). Please find the results at https://ibb.co/eLJqfV .
> > >
> > > Most interestingly, for certain binarization thresholds the top-5 accuracy is within 1-2% of the vanilla BagNet performance. This indeed supports your intuition that the amplitude of the heatmaps is not the most important factor. We will include these results (with some more intermediate values) in the appendix.

---

> > > > ### Comment · AnonReviewer1 · 2018-11-30
> > > > **Thanks for this rebuttal**
> > > >
> > > > Thank you. It answered all my concerns/questions.

---

### Official Review · AnonReviewer3 · 2018-11-03
**This paper is worth being accepted. The bag-of-words information in the neural network is important for high prediction accuracy. Possibly has high impact in the community and need to be further investigated.**

**Rating:** 7
**Confidence:** 4

**Review:**

This paper suggests a novel and compact neural network architecture which uses the information within bag-of-words features. The proposed algorithm only uses the patch information independently and performs majority voting using independently classified patches. The proposed method provides the state-of-the-art prediction accuracy unexpectedly, and several additional experiments show the state-of-the-art neural networks mainly learn without association between information in different patches.

The proposed algorithm is simple and does not provide completely new idea, but this paper has a clear contribution connecting the previous main idea of feature extraction, bag-of-words, and the prevailing blackbox algorithm, CNN. The results in the paper are worth to be shared in the community and need further investigated.

The presented experiments look fair and reasonable to show the importance of the independent patch information (without association between them), and the presented experimental results show some state-of-the-art methods also perform with independent patch information.

Comparison with attention models is necessary to compare the important patches obtained from conventional networks.

---

> ### Author Response · Authors · 2018-11-22
> **Thanks!**
>
> Thanks a lot for appreciating our contribution!
>
> > Comparison with attention models is necessary to compare the important patches obtained from conventional networks.
>
> In the paper  (section 4.3) we quantitatively show that the patches important to BagNets are also important for standard CNNs. Is that the direction you were thinking about? If you have a different experiment in mind we would like to kindly ask you for more details.

---

### Official Review · AnonReviewer2 · 2018-11-05
**Combing Patch-level CNN and BoF model has been done before, but the paper has the smallest patch**

**Rating:** 6
**Confidence:** 4

**Review:**

The idea of image classification based on patch-level deep feature in the BoF model has been studied extensively.

 Just list few of them:

Wei et al. HCP: A Flexible CNN Framework for Multi-label Image Classification, IEEE TPAMI 2016
Tang et al. Deep Patch Learning for Weakly Supervised Object Classification and Discovery, Pattern Recognition 2017
Tang et al. Deep FisherNet for Object Classification, IEEE TNNLS
Arandjelović et al. NetVLAD: CNN Architecture for Weakly Supervised Place Recognition, CVPR 2016

The above papers are not cited in this paper.

There are some unique points. This work does not use RoIPooling layer and has results on ImageNet. But, the previous works use RoIPooling layer to save computations and works on scene understanding images, such as PASCAL.

Besides, the paper uses the smallest patch among all the patch-based deep networks. It is interesting.

I highly encourage the authors to finetune the ImageNet pre-trained BagNet on PASCAL VOC and compare to the previous patch-based deep networks.

---

> ### Author Response · Authors · 2018-11-06
> **We do cite similar approaches but they use whole-object patches (instead of small parts), barely increase interpretability and do not shed light on decision making in CNNs**
>
> Thank you for reviewing our paper. We would like to make a quick clarification right away, which we hope will change your assessment. All works you cite use non-linear BoF encodings on top of pretrained VGG (or AlexNet) features; the effective patch size of individual features is thus large and will generally encompass the whole object of interest. In contrast, our BagNets are constrained to very small image patches (much smaller than the typical object size in ImageNet), use no region proposals (all patches are treated equally) and employ a very simple and transparent average pooling of local features (no non-linear dependence between features and regions). That’s why BagNets (1) substantially increase interpretability of the decision making process (see e.g. heatmaps), (2) highlight what features and length-scales are necessary for object recognition and (3) shed light on the classification strategy followed by modern high performance CNNs. None of the cited papers addresses any of these contributions.
>
> PS: We do cite similar approaches in our paper, see first paragraph of related literature. We will add your references there.

---

> > ### Comment · AnonReviewer2 · 2018-11-06
> > **I still have the previous questions**
> >
> > Thanks for the authors' response!
> >
> > What are the similar papers cited in the paper?
> >
> > In the previous patch-based deep learning methods, there are multi-scale patches. For example, in PASCAL VOC, the whole image is about 500*600 px and the small patch is 32*32 px; they are not whole-object patches. In fact, it is not impossible to obtain whole-object patches, unless object detection has been perfectly done :)
> >
> > Regarding the effectiveness of highlighting the useful features/patches to explain CNNs, this also has been done before. Please refer to the papers I mentioned before; there are figures to useful patches. In computer vision, there are many papers working on learning mid-level features, meaningful patterns or deep patterns. You may also refer to them.
> >
> > In my understanding, methodologically, there is nothing new in the paper. The explanations about the interpretability of deep nets are not deep enough (not inside of the deep net) and there are many works had ready done similar things.
> >
> > Besides, the time complexity issue of BagNet is not addressed in the paper.

---

> > > ### Comment · AnonReviewer1 · 2018-11-06
> > > **-**
> > >
> > > @R2: can you comment on the receptive field size of the final layer of the BagNet versus the works you mentioned?

---

> > > > ### Comment · AnonReviewer2 · 2018-11-07
> > > > **I agree with the authors that BagNets have the smallest patches and the other papers (Tang et al and the NetVLAD) papers have "large patch" due to the receptive fields.**
> > > >
> > > > I agree with the authors that BagNets have the smallest patches and the other papers (Tang et al and the NetVLAD) papers have "large patch" due to the receptive fields.
> > > >
> > > > The first work in this field [3] uses raw patches and does not have receptive fields.
> > > >
> > > > [3] Yunchao Gong, Liwei Wang, Ruiqi Guo, and Svetlana Lazebnik. Multi-scale orderless pooling of deep convolutional activation features.
> > > >
> > > > However, [3] is not end-to-end trained.
> > > >
> > > > So the way of combining CNN with BoF is different from the previous works. But it is not fundamentally different.
> > > >
> > > > If all the rest reviewers are willing to accept the paper, I can give a weak accept.
> > > >
> > > > But, still, I want the authors to give more details about the time complexity and speed. In addition, to make the paper more convincing, the authors should use RoIPooling to compare with [Tang].
> > > >
> > > > [Tang] Tang et al. Deep Patch Learning for Weakly Supervised Object Classification and Discovery, Pattern Recognition 2017

---

> > > > > ### Author Response · Authors · 2018-11-07
> > > > > **Thanks for the open discussion and further comments**
> > > > >
> > > > > First and foremost thanks for this open debate and for reconsidering your decision. We will try to clarify the relation to the works you mentioned in our manuscript. A few more comments from our side:
> > > > >
> > > > > (1) Reference [3] builds a new feature vector for an image that combines a feature vector of the whole image with feature vectors of 128 x 128 and 64 x 64 pixel patches (in order to increase invariance to image transformations). The resulting classification is thus neither more interpretable nor constrained to small patches.
> > > > >
> > > > > (2) You mention that we should use RoIPooling to compare with [Tang]. On what metric would you want this comparison to be performed? We do not claim performance advantages in terms of object classification and do not perform object discovery (one of the subgoals of [Tang et al] which is why they use PASCAL VOC). We'd very much appreciate if you could clarify what exact experiment you have in mind.
> > > > >
> > > > > (3) Time complexity of BagNets: a BagNet-32/16/8 running on a GTX 1080 Ti can process 155 (+- 5) images / second of size 3 x 224 x 224 (in batches of 10). For comparison, a ResNet50 can process 570 images in the same time, so BagNets are around 3 to 4 times slower than standard ResNets. Please remember that BagNets are basically ResNets but with most 3x3 convolutions replaced by 1x1 convolutions, so this timing is roughly expected (we have less spatial dimensionality reduction which explains the increased runtimes).
> > > > >
> > > > > All in all, the main contributions of this work are (1) a more transparent and interpretable object recognition pipeline (in terms of precisely which object features are being used for classification), (2) the insight that ImageNet can be solved to high accuracy with very small and local image features (so e.g. no shape recognition is required to solve ImageNet) and (3) the insight that standard and widely used ImageNet CNNs seem to use a similar BoF classification strategy. We believe that these insights go way beyond previous work and are not at all addressed in the region proposal literature. Please note that we do not want to claim that our architecture is revolutionary but that we can draw important insights from it about object classification in natural images and what internal decision making process CNNs may use in these tasks (which, given the lack of understanding of current CNN architectures, is dearly needed).

---

> > > > > > ### Comment · AnonReviewer1 · 2018-11-07
> > > > > > **-**
> > > > > >
> > > > > > @authors: I'm not sure you design a more interpretable CNN: your analysis is purely spatial. I think this should be weakened in the writing because it is misleading. I agree with the other points otherwise.

---

> > > > > > > ### Author Response · Authors · 2018-11-07
> > > > > > > **Thanks for your feedback**
> > > > > > >
> > > > > > > Thanks for your feedback! The word "interpretable" has different meanings for different people and we agree that we should be careful to define exactly what we mean by this term. There is a large body of literature trying to "understand" CNN decisions by means of a post-hoc feature attribution (i.e. which image parts have been important for the decision). So we mean "more interpretable" in the sense that this architecture transparently integrates evidence from different spatial locations and thus lets us precisely track which spatial features have been contributing how much to the final decision. For each individual location, however, the CNN feature extraction is still a black box. In other words, we reduced the complexity (and thus increased interpretability) of CNN decision making by introducing a transparent and interpretable spatial aggregation mechanism on top of a (still blackbox) local feature extraction. We'll update the manuscript to reflect this perspective more clearly and would appreciate your feedback.

---

> > > > > > > > ### Comment · AnonReviewer1 · 2018-12-17
> > > > > > > > **-**
> > > > > > > >
> > > > > > > > Dear author,
> > > > > > > >
> > > > > > > > I guess I missed this answer. I'm not sure it is fair to claim this CNN is more interpretable, in the sens that this work opens more questions than it closes. "a transparent and interpretable spatial aggregation mechanism", is a bit of an overkill in my humble opinion. Do not worry, this does not affect my review or score, however I do prefer to be honest on this point.
> > > > > > > >
> > > > > > > > I think in the current context of DL, such works should make clear statements of their contributions(and there are a lot here!)
> > > > > > > >
> > > > > > > > Regards,
> > > > > > > > _

---

> > > > > > > > > ### Author Response · Authors · 2018-12-21
> > > > > > > > > **Thanks for your thoughts**
> > > > > > > > >
> > > > > > > > > Thanks for your thoughts and honest opinion! The decision period might be over but I'd still be curious to get a better understanding of your point of view. To be concrete, in the TL;DR we formulated:
> > > > > > > > >
> > > > > > > > > "Aggregating class evidence from many small image patches suffices to solve ImageNet, yields more interpretable models and can explain aspects of the decision-making of popular DNNs."
> > > > > > > > >
> > > > > > > > > You seem to weigh the contributions of our paper a bit differently. If you are in the mood and can spare the time I would be very grateful if you could formulate an alternative TL;DR that is better aligned with your perception of the work.

---

> > > > > ### Comment · AnonReviewer1 · 2018-11-07
> > > > > **-**
> > > > >
> > > > > @R2: "[3]" can you comment on the accuracy of the paper you report?
> > > > >
> > > > > @R2, "time complexity and speed": Do you think it would be possible to design cuda routines that act in parallel on patches?
> > > > > However, I agree the memory use is more tricky, but I'm ok with it; this is not an engineering paper.
> > > > >
> > > > > @R2: "ROIPooling": could you point us to a paper using it for classification? I'd be very interested to read more about it. Thanks.

---

> > > > > ### Author Response · Authors · 2018-11-22
> > > > > **-**
> > > > >
> > > > > Thanks again for the open discussion and for thinking about raising your score. Please let us know if there are any further clarifications or questions that we should address.

---

> > > ### Author Response · Authors · 2018-11-06
> > > **The effective minimum patch size in the cited works is much larger than 32 x 32 pixels**
> > >
> > > Thanks for taking the time to respond! To be concrete we'll refer to Tang et al. 2017 in our response.
> > >
> > > We believe the statement that "the small patch is 32x32 pixels" is based on a confusion between region proposals (the patches/bounding boxes that you see) and receptive fields. The region proposals spatially crop parts of the highest conv layer activations (e.g. for VLAD encoding, see Figure 4 in Tang et al.). What is shown in visualisations is the image part that corresponds to the cropped part (i.e. if 1/4 of the conv layer is cropped then 1/4th of the image is shown as proposal region). But that is misleading: since each feature vector already sees large parts of the image (212 x 212 pixels in VGG16 to be precise), the effective image region is much larger then the visualised region proposal (minimum is 212 x 212 pixels).
> > >
> > > > Besides, the time complexity issue of BagNet is not addressed in the paper.
> > >
> > > BagNets have roughly the same runtime as standard ResNet-50's (it's slightly higher because we have less pooling). We will add precise measurements to the paper, thanks for the suggestion.
> > >
> > > As for previous work, in the corresponding section we wrote "Predominantly, DNNs were used to replace the previously hand-tuned feature extraction stage in BoF models, often using intermediate or higher layer features of pretrained DNNs" which, as far as we can see, pretty much applies to the paper you cite (all of them are based on high layer features of AlexNet and VGG). The references that we cite are:
> > >
> > > [1] Jiewei Cao, Zi Huang, and Heng Tao Shen. Local deep descriptors in bag-of-words for image retrieval. In Proceedings of the on Thematic Workshops of ACM Multimedia 2017
> > > [2] Jiangfan Feng, Yuanyuan Liu, and Lin Wu. Bag of visual words model with deep spatial features for geographical scene classification. Comp. Int. and Neurosc., 2017:5169675:1–5169675:14, 2017.
> > > [3] Yunchao Gong, Liwei Wang, Ruiqi Guo, and Svetlana Lazebnik. Multi-scale orderless pooling of deep convolutional activation features.
> > > [4] Eva Mohedano, Kevin McGuinness, Noel E. O’Connor, Amaia Salvador, Ferran Marques, and Xavier Giro-i Nieto. Bags of local convolutional features for scalable instance search. In Proceedings of the 2016 ACM on International Conference on Multimedia Retrieval, ICMR ’16,
> > > [5] Joe Yue-Hei Ng, Fan Yang, and Larry S. Davis. Exploiting local features from deep networks for image retrieval. In CVPR Workshops,
> > > [6] Fahad Shahbaz Khan, Joost van de Weijer, Rao Muhammad Anwer, Andrew D. Bagdanov, Michael Felsberg, and Jorma Laaksonen. Scale coding bag of deep features for human attribute and action recognition. CoRR, abs/1612.04884, 2016.

---

> ### Author Response · Authors · 2018-11-06
> **Another perspective that might help**
>
> Maybe the following perspective also helps: the works you cite use BoF over larger image regions, but the embeddings for each region are still based on conventional, non-interpretable DNNs (like VGG). Our work "opens this blackbox" (to use a very stressed term) and provides a way to compute similar region embeddings in a much more interpretable way as a linear BoF over small patches. In other words, if the works you cite would use BagNets instead of VGG, they would basically use a "stacked BoF" approach: first, small and local patches are combined to yield region embeddings (BagNet), and these region embeddings are used by a second BoF to infer image-level object labels and bounding boxes.

---

### Public Comment · ~Eugene_Belilovsky1 · 2018-10-15
**Interesting work and insights, a potentially related reference**

This paper was a nice read. I find the results in this paper quite interesting and it is refreshing to see work revisiting some of the underlying assumptions in our modern computer vision pipelines. I wanted to point out a  related result in our recent work https://arxiv.org/pdf/1703.08961.pdf (Sec 2.3,3.1) where we show that using a model localized 16x16 patches can obtain an AlexNet accuracy on imagenet. Specifically we had used a (non-overlapping) local transform with a 16x16 window followed by 3 1x1 convolutions and then an MLP. Indeed, although the MLP could potentially exploit more global spatial information we conjectured this would be quite hard/unlikely, and I believe your result that directly aggregates the predictions of the local encodings reaching nearly the same accuracy confirms this to a degree.

I was also wondering if you have tested models other than resnet50-like models  as your base, and if so whether those gave substantial differences in the result when varying the actual model  (e.g. shallower/thinner/ or non-residual). One could speculate that models applied to smaller sized patches could require a less complex network than is typically used (a potential advantage of this approach). If I understood your model is already rather small compare to the base resnet-50?

---

> ### Author Response · Authors · 2018-10-29
> **Architecture search could increase performance and/or efficiency**
>
> Dear Eugene,
> thanks for your comment and the interesting reference! Indeed our results seem to confirm your suspicions regarding integration of spatial information. We'll add your paper into our related work section.
>
> We did not vary our base architecture. For those reasons I'd expect that one can reach even higher performance with a suitable hyperparameter/architecture search or be much more efficient (more shallow/thin architecture) than what we currently use. These could definitely be interesting future directions to pursue.

---

### Public Comment · ~Seunghyeon_Kim1 · 2018-11-05
**Interesting idea and results with some comments**

This paper combines the concept of Bag-of-Feature (BoF) with modern DNN to propose more interpretable neural network framework. Since the proposed method can achieve similar performance to the modern DNN, it can be an alternative to DNN. However, the paper lacks a description of the test phase so it is not clear how many qxq patches are extracted from the full image. As I understand, BagNet extract many small patches from the image, so probably it takes a long time to test one image. In my opinion, it is good to report the test time for the image.

The most interesting part of this paper is section 4.3  which supports the argument that modern DNN learns similar local features to the BagNet. The four experiments in section 4.3 show that VGG16 acts quite similar to the BagNet. On the other hand, the same experiments clearly show that deeper networks such as ResNet-51, DenseNet act totally different from BagNet. In my opinion, these results seem to be contrary to the contribution of the paper that modern DNN can be explained as BoF framework.

---

> ### Author Response · Authors · 2018-11-22
> **Runtime is fast and deeper networks are only gradually shifting away from linear BagNets**
>
> Thanks for your comments! Regarding runtimes, a BagNet-32/16/8 running on a GTX 1080 Ti can process 155 (+- 5) images / second of size 3 x 224 x 224 (in batches of 10). For comparison, a ResNet50 can process 570 images in the same time, so BagNets are around 3 to 4 times slower than standard ResNets. Please remember that BagNets are basically ResNets but with most 3x3 convolutions replaced by 1x1 convolutions, so this timing is roughly expected (we have less spatial dimensionality reduction which explains the increased runtimes).
>
> As for deeper neural networks, we indeed observe a large deviation from BagNets the deeper and more precise the networks are. This can be observed from stronger non-linear interactions between spatial patches (Figure 6) and the reduced effectiveness of masking local regions (Figure 8). These deviations may come from (1) a more non-linear classifier on top of the local features or (2) larger local feature sizes. In any case, this is a gradual shift away from linear BagNets and we see it as a refinement of our results, not a contradiction.

---

### Author Response · Authors · 2018-11-26
**Author's summary of rebuttal discussion**

We would like to thank all reviewers for their valuable feedback and we very much appreciate their assessment of our work as “interesting” (R1), “worth to be shared in the community” (R2) and “a valuable contribution to ICLR” (R3).

We responded in detail to the comments of each reviewer below and summarise here the main changes to the manuscript:

* We clarified that this work is unrelated to region proposal models (like Wei et al. or Tang et al.) in the related work section. (R1)
* We added an experiment probing the sensitivity of the BagNets to the precise numerical values of the heatmaps in the appendix. (R3)
* We added runtime measurements for BagNets to the results section. (R1)
* We added a paragraph in the introduction to define precisely our meaning of interpretability. (R3)
* We defined our notion of “linear BoF models” and commented on its relevance in the model description. (R3)
* We straightened the use of the word “feature" throughout the manuscript. (R3)
* We modified Figure A.1 to clarify the downsampling step. (R3)

---

### Meta-Review · Area_Chair1 · 2018-12-13

**Confidence:** 4
**Recommendation:** Accept (Poster)

**Metareview:**

This paper presents an approach that relies on DNNs and bags of features that are fed into them, towards object recognition.  The strength of the papers lie in the strong performance of these simple and interpretable models compared to more complex architectures.  The authors stress on the interpretability of the results that is indeed a strength of this paper.

There is plenty of discussion between the first reviewer and the authors regarding the novelty of the work as the former point out to several related papers;  however, the authors provide relatively convincing rebuttal of the concerns.

Overall, after the long discussion, there is enough consensus for this paper to be accepted to the conference.